# In Vitro Mineralisation of Tissue-Engineered Cartilage Reduces Endothelial Cell Migration, Proliferation and Tube Formation

**DOI:** 10.3390/cells12081202

**Published:** 2023-04-20

**Authors:** Encheng Ji, Lieke Leijsten, Janneke Witte-Bouma, Adelin Rouchon, Nunzia Di Maggio, Andrea Banfi, Gerjo J. V. M. van Osch, Eric Farrell, Andrea Lolli

**Affiliations:** 1Department of Oral and Maxillofacial Surgery, Erasmus MC University Medical Center Rotterdam, 3015 GD Rotterdam, The Netherlands; 2Department of Biomedicine, Basel University Hospital, University of Basel, 4031 Basel, Switzerland; 3Department of Orthopaedics and Sports Medicine, Erasmus MC University Medical Center Rotterdam, 3015 GD Rotterdam, The Netherlands; 4Department of Otorhinolaryngology, Erasmus MC University Medical Center Rotterdam, 3015 GD Rotterdam, The Netherlands; 5Department of Biomechanical Engineering, University of Technology Delft, 2628 CD Delft, The Netherlands

**Keywords:** bone tissue engineering, endochondral ossification, angiogenesis, mineralisation, mesenchymal stromal cells

## Abstract

Tissue engineering bone via endochondral ossification requires the generation of a cartilage template which undergoes vascularisation and remodelling. While this is a promising route for bone repair, achieving effective cartilage vascularisation remains a challenge. Here, we investigated how mineralisation of tissue-engineered cartilage affects its pro-angiogenic potential. To generate in vitro mineralised cartilage, human mesenchymal stromal cell (hMSC)-derived chondrogenic pellets were treated with β-glycerophosphate (BGP). After optimising this approach, we characterised the changes in matrix components and pro-angiogenic factors by gene expression analysis, histology and ELISA. Human umbilical vein endothelial cells (HUVECs) were exposed to pellet-derived conditioned media, and migration, proliferation and tube formation were assessed. We established a reliable strategy to induce in vitro cartilage mineralisation, whereby hMSC pellets are chondrogenically primed with TGF-β for 2 weeks and BGP is added from week 2 of culture. Cartilage mineralisation determines loss of glycosaminoglycans, reduced expression but not protein abundance of collagen II and X, and decreased VEGFA production. Finally, the conditioned medium from mineralised pellets showed a reduced ability to stimulate endothelial cell migration, proliferation and tube formation. The pro-angiogenic potential of transient cartilage is thus stage-dependent, and this aspect must be carefully considered in the design of bone tissue engineering strategies.

## 1. Introduction

Endochondral ossification is the most common process of bone formation and long bone elongation. It involves the condensation of mesenchymal cells which form a cartilaginous template of the future bone. Over time, these cells become enlarged (hypertrophy), and the cartilage matrix is mineralised and remodelled. Pro-angiogenic factors secreted by hypertrophic chondrocytes lead to blood vessel invasion, which allows the homing of remodelling osteoclasts and bone-forming osteoblasts [1,2,3]. The endochondral ossification process has been reproduced in the tissue engineering setting by several research groups around the world as a promising strategy to (re)generate bone [4,5,6,7,8]. This involves priming cells chondrogenically in vitro, followed by implantation in rodents to allow mineralisation, remodelling, vascularisation and formation of the true bone organ to occur. To date, this approach has not yet been translated to the patient, partly due to the effort required to expand and prime the cells in vitro, as well as the long time necessary to achieve complete remodelling of the implanted cartilage into mature bone following implantation. Improving these aspects remains a significant challenge, particularly since our understanding of the kinetics of cartilage vascularisation and remodelling in the tissue engineering setting is still limited. In vitro models of the early stages of endochondral ossification are lacking, further limiting our ability to study how hypertrophic cartilage is remodelled and to elucidate the underlying crosstalk between the several cell types that take part in the process.

The most commonly used cell type for bone tissue engineering is the marrow stromal cell (MSC), which is differentiated chondrogenically for as little as 7 days [9] to as many as 42 days [10]. After implantation of MSC-derived cartilage in an animal, rapid mineralisation occurs (within 7 days, unpublished data). In our previous study, we proposed that an adequate chondrogenic matrix and hypertrophic chondrocytes are the requirements for successful mineralisation and consequent bone formation [9]. However, vascularisation of the (mineralised) cartilage, which allows the homing of the cell types that can drive cartilage remodelling, is observed only much later (4 to 8 weeks post-implantation), likely due to slow invasion of a dense and compact matrix. Achieving sufficient and rapid implant vascularisation is a general concern in the bone tissue engineering field, and various strategies have been proposed to improve vascularisation dynamics, such as incorporating angiogenic growth factors [11,12] and including hollow channels in the constructs [13,14,15]. Alternatively, several studies have investigated the use of pre-vascularisation strategies [16,17,18], whereby a rudimentary microvasculature pre-formed in vitro undergoes anastomosis with host blood vessels after implantation. A better understanding of the interactions between vessel-forming (endothelial) cells and bone-forming constructs is required for improved strategies to (pre-)vascularise tissue-engineered implants in vitro or in vivo [19,20]. Particularly in the field of endochondral tissue engineering, research is needed to identify and harness the optimal cues that could drive the rapid vascularisation of chondrogenic implants.

Angiogenesis, the formation of new blood vessels from pre-existing vessel structures, provides the nutrients required for bone formation and growth, and for cells involved in bone formation such as osteoblasts and monocytes/osteoclasts. During endochondral ossification, several cell types including hypertrophic chondrocytes [21], osteoclasts [22] and septoclasts [23] produce catabolic enzymes which mediate extracellular matrix (ECM) proteolysis [24] and pave the way for vessel infiltration [25,26]. The endothelial cells themselves secrete matrix metalloproteinases (MMPs), such as MMP2 and MMP9, to facilitate vessel ingrowth [27]. Much research into the function of angiogenesis and its modulators in endochondral ossification has been conducted, but mainly in connection to the production of pro-angiogenic and anti-angiogenic factors by hypertrophic cartilage [28,29,30]. Previous work from our group and others has demonstrated that both human patient-derived cartilage and tissue-engineered cartilage (chondrogenic hMSC pellets) are pro-angiogenic [31,32,33,34]. Nevertheless, how cartilage mineralisation may impact the pro-angiogenic potential of cartilage has not been broadly investigated. Increased concentrations of calcium and phosphate have been shown to increase oxidative stress [35] and apoptosis [36] in endothelial cells. Therefore, we hypothesise that the mineralisation of MSC-derived cartilage could impact its ability to attract endothelial cells and stimulate the formation a vascular network. Importantly, this may strongly affect the vascularisation of chondrogenic implants, which undergo rapid mineralisation after implantation and prior to or during vascular infiltration. Furthermore, understanding the interactions between mineralised cartilage and endothelial cells will be crucial for tissue engineering strategies which aim to induce cartilage mineralisation prior to implantation to accelerate in vivo bone formation.

In this work, we aimed to investigate how in vitro mineralisation of MSC-derived cartilage affects its ability to stimulate endothelial cell migration, proliferation and tube formation. First, we aimed to setup an optimised strategy to induce the in vitro mineralisation of chondrogenically primed hMSC pellets. We then characterised the response of the pellets to mineralisation in terms of matrix changes and expression of angiogenic markers. Finally, we examined how cartilage mineralisation affected endothelial cell behaviour and the formation of microvascular networks in vitro.

## 2. Materials and Methods

### 2.1. Cell Culture

Human bone marrow MSCs were used to generate chondrogenic and mineralised pellets (N = 5 donors in total). Donors 1–2 were used to optimise the protocol of in vitro mineralisation. Donors 3–5 were subjected to a time-course analysis to study the effect of mineralisation on the ability of tissue-engineered cartilage to stimulate endothelial cell migration, proliferation and tube formation. The cells were isolated from leftover materials obtained from paediatric patients undergoing alveolar bone graft surgery (5 male patients, age 9–12 years). All samples were harvested after informed consent and with the approval of the Medical Ethics Review Committee at Erasmus MC University Medical Center Rotterdam, The Netherlands (MEC2014-106). Cells were plated in alpha minimum essential medium (αMEM, ThermoFisher, Waltham, MA, USA) containing 10% *v*/*v* heat inactivated foetal bovine serum (FBS, Sigma-Aldrich, St. Louis, MO, USA) and supplemented with 50 µg/mL gentamycin (ThermoFisher, Waltham, MA, USA), 1.5 µg/mL Amphotericin B (ThermoFisher, Waltham, MA, USA), 25 µg/mL ascorbic acid 2-phosphate (Sigma-Aldrich, St. Louis, MO, USA), 1 ng/mL fibroblast growth factor-2 (Instruchemie B.V., Delfzijl, The Netherlands) and fungizone/gentamycine (F/G, ThermoFisher, Waltham, MA, USA), with a seeding density of 2300 cells/cm^2^, in T175 flasks. After 24 h, flasks were washed to remove non-adherent cells and debris, and the medium was renewed. Cells were cultured in a humidified atmosphere at 37 °C and 5% carbon dioxide (CO_2_), and the medium was changed twice a week. hMSCs were subcultured once 85–90% confluence was reached, using 0.05% *w*/*v* trypsin (ThermoFisher, Waltham, MA, USA), and replated at a density of 2300 cells/cm^2^. Cells were expanded until passage 2–3 for pellet formation.

Adipose-derived stromal/stem cells (ASCs) were co-cultured with endothelial cells for the generation of a 3D vascular network. The cells were isolated from human adipose tissue obtained from one healthy patient undergoing plastic surgery after informed consent and approval from the Ethical Committee of the Basel University Hospital (Ethikkommission beider Basel [EKKB], Ref. 78/07). The adipose tissue was minced and digested with 0.15% *w*/*v* collagenase (Worthington Biochemical Corporation, Lakewood, USA) in phosphate-buffered saline (PBS) at 37 °C under continuous shaking for 60 min. After centrifugation at 1500 rpm for 10 min, the lipid-rich layer was discarded and the cellular pellet was washed once with PBS. Released cells were strained through a 100-μm strainer to remove fibrous debris. Cells were plated in T175 flasks at the density of 10,000 cells/cm^2^. High-glucose Dulbecco’s modified Eagle’s medium (ThermoFisher, Waltham, MA, USA) with 10% *v*/*v* heat inactivated foetal bovine serum (Sigma-Aldrich, St. Louis, MO, USA), 1% HEPES (ThermoFisher, Waltham, MA, USA), 1% glutamine solutions, 1% penicillin/streptomycin (P/S, ThermoFisher, Waltham, MA, USA) and 5 ng/mL fibroblast growth factor-2 (FGF-2, R&D System, Minneapolis, MN, USA) were used for cell culture. Cells were expanded until passage 1 for experiments.

Human umbilical vein endothelial cells (HUVECs) were purchased from Promocell (C-12203; pooled donors) and expanded until passage 4 with a seeding density of nearly 3300 cells/cm^2^. Endothelial cell growth medium-2 (EGM2, Promocell, Huissen, The Netherlands or Lonza, Basel, Switzerland) was used for HUVEC expansion. The medium was changed twice a week. Cells were subcultured when 85–90% confluence was reached, until passage 4 or 5.

### 2.2. Generation and In Vitro Differentiation of hMSC Pellets

200,000 hMSCs were resuspended in 500 μL of complete chondrogenic medium [high-glucose Dulbecco’s modified Eagle’s medium (DMEM) supplemented with 50 µg/mL gentamycin (ThermoFisher, Waltham, MA, USA), 1.5 µg/mL fungizone (ThermoFisher, Waltham, MA, USA), 1 mM sodium pyruvate (ThermoFisher, Waltham, MA, USA), 40 µg/mL proline (Sigma-Aldrich, St. Louis, MO, USA), 1:100 *v*/*v* insulin-transferrin-selenium (ITS+; BD Biosciences, Franklin Lakes, NJ, USA), 10 ng/mL transforming growth factor-β3 (TGF-β3, R&D System, Minneapolis, MN, USA), 25 µg/mL L-ascorbic acid 2-phosphate (Sigma-Aldrich, St. Louis, MO, USA) and 100 nM dexamethasone (Sigma-Aldrich, St. Louis, MO, USA)] in 15 mL-polypropylene tubes and centrifuged for 8 min at 200× *g*. After 24 h of culture, the tubes were gently tapped to dislodge the pellets, which were further cultured in a humidified atmosphere at 37 °C in 5% CO_2_ for up to 28 days. To induce mineralisation of chondrogenic pellets, 10 mM β-Glycerophosphate (BGP, Sigma-Aldrich, St. Louis, MO, USA) was added to the culture medium, according to the different culture schemes reported in Figure 1a.

### 2.3. Calcium Uptake Assay

The mineralisation of chondrogenic pellets was monitored during culture by determining the calcium uptake from the medium. To monitor calcium uptake by the pellets, 100 μL of supernatant was collected from 3 pellets for each condition twice a week, and the calcium concentration was calculated using a standard curve of 0–3.0 mM CaCl_2_ (Sigma-Aldrich, St. Louis, MO, USA) in calcium-free DMEM (ThermoFisher, Waltham, MA, USA). In a 96-well plate, 100 μL of reagent [1:1 of reagent 1 (1M ethanolamine pH 10.5 (Sigma-Aldrich, St. Louis, MO, USA)) and reagent 2 (0.35 mM o-cresolphthalein complexone (Sigma-Aldrich, St. Louis, MO, USA), and 19.8 mM 8-hydroxyquinoline (Sigma-Aldrich, St. Louis, MO, USA), 0.6 M hydrochloric acid)] were added to 10 μL medium or standard. The assay is based on the reaction between Ca^2+^ in the culture medium, and o-Cresophthalein complexone in an alkaline solution. This produces a purple-pink colour that was measured at 570 nm on a Versamax spectrophotometer. A standard curve generated with samples with known Ca^2+^ concentration (standards) was used to calculate the concentration of experimental samples by interpolation. Samples consisting of medium only (no pellets) were taken during culture and used as a blank for data normalisation.

### 2.4. Generation of Conditioned Medium from hMSC Pellets

At time points 7, 14, 21 and 28 days, the culture medium of pellets was renewed. Then, 24 h later, the pellets were washed with PBS three times and incubated with basal medium (high-glucose DMEM supplemented with 50 µg/mL gentamycin (Thermofisher, Waltham, MA, USA), 1.5 µg/mL fungizone (Thermofisher, Waltham, MA, USA), 0.1% *w*/*v* bovine serum albumin (Sigma-Aldrich, St. Louis, MO, USA), and 0.1 mM L-ascorbic acid 2-phosphate (Sigma-Aldrich, St. Louis, MO, USA) for 24 h (37 °C and 5% CO₂). After 24 h, the conditioned medium (CM) of 16 pellets per condition was collected and pooled, and cell debris was removed by centrifugation at 700× *g* for 8 min at 4 °C. The same procedure was performed with basal medium without pellets to generate a non-conditioned medium (negative control). All the media were stored at −80 °C until use.

### 2.5. Live/Dead Flow Cytometry Assay

To evaluate cell viability, pellets were collected for live/dead fluorescence-activated cell sorting (FACS) assay. hMSC pellets were incubated with 3 mg/mL collagenase A (Sigma-Aldrich, St. Louis, MO, USA) and 1.5 mg DNase I (Sigma-Aldrich, St. Louis, MO, USA) in RPMI-1640 media (ThermoFisher, Waltham, MA, USA) containing 5% FBS, at 37 °C for 90 min. After incubation, the cell suspension was filtered through a 100-µm cell strainer to remove the pellet debris, and centrifuged at 400× *g* for 5 min. One vial of fluorescent reactive dye (Component A) and 50 μL of anhydrous DMSO (Component B) from the LIVE/DEAD™ Fixable Dead Cell Stain Kit (ThermoFisher, Waltham, MA, USA) were mixed. PBS was used to wash and resuspend cells, and cell numbers were adjusted to the density of 1 × 10^6^ cells/mL by cell counting and PBS. 1 μL of the reconstituted fluorescent reactive dye was mixed with 1 mL of the cell suspension. After 30 min incubation in the dark, cells were analysed with a FACS Jazz cell sorter (Becton Dickinson, Franklin Lakes, NJ, USA), and the results were processed with FlowJo software version 10.0.7 (FlowJo LLC, Ashland, Wilmington, DE, USA).

### 2.6. Histological Analysis

hMSC pellets were fixed overnight in 4% formalin prior to dehydration and paraffin-wax embedding. Sections 6 μm thick were cut from all samples. Slides were deparaffinised with xylene and then rehydrated with ethanol gradients. Cell morphology was assessed by haematoxylin and eosin (H&E) staining. To evaluate mineralisation, von Kossa staining was performed. Slides were washed with distilled water and immersed in a silver nitrate solution (Sigma-Aldrich, St. Louis, MO, USA) for 10 min under a desk light (>60 W). Then, 5% sodium thiosulphate (Sigma-Aldrich, St. Louis, MO, USA) was used to remove unreacted silver nitrate. Finally, the slides were mounted with mounting solution (VectaMount, Vector Laboratories, Newark, NJ, USA) and enclosed with coverslips. The staining was quantified by a computerised video camera-based image analysis system (NIH, USA ImageJ software, public domain available at: http://rsb.info.nih.gov/nih-image/ (accessed on 17 April 2023)) under brightfield microscopy and expressed as % of mineralised area (3 sections/pellet at different depths). For the evaluation of matrix glycosaminoglycan (GAG), thionine staining was performed. Deparaffinised slides were stained with 0.4% thionine (Sigma-Aldrich, St. Louis, MO, USA) in 0.01 M aqueous sodium acetate (Sigma-Aldrich, St. Louis, MO, USA), pH 4.5, for 5 min. Then, slides were immersed in 70% ethanol (10 s), 96% ethanol (30 s), 100% ethanol (1 min), and xylene (twice for 1 min) for differentiating the staining. Finally, the slides were mounted with mounting solution (VectaMount, Vector Laboratories, Newark, NJ, USA) and coverslips.

Immunocytochemistry for collagen type II and X was employed to evaluate collagens in the matrix of the pellets. For antigen retrieval, slides were treated with 0.1% pronase (Sigma-Aldrich, St. Louis, MO, USA) at 37 °C for 30 min for collagen type II; for collagen type X, pepsin 1 mg/mL in 0.5 M acetic acid pH 2 for 2 h at 37 °C was used for this step. Afterwards, for both types of staining, 10 mg/mL hyaluronidase (Sigma-Aldrich, St. Louis, MO, USA) in PBS was applied to improve antibody penetration, at 37 °C for 30 min. Then, samples were incubated with 10% normal goat serum (Southern Biotech, Birmingham, USA) in PBS with 1% BSA (ThermoFisher, Waltham, MA, USA). Slides were subsequently incubated with either mouse monoclonal 1:100 1st antibody against collagen type II (DSHB, 0.4 μg/mL stock) or 1:100 collagen type X (5 μg/mL stock, ThermoFisher, Waltham, MA, USA) overnight. The slides were then incubated with a biotinylated 1:100 goat-anti-mouse antibody (Biogenex, Fremont, CA, USA) for 30 min followed by an incubation with 1:50 streptavidin-AP (Biogenex, Fremont, CA, USA). Staining was revealed by incubation with New Fuchsin substrates (Chroma, 1 g/25 mL with 2M HCl). Finally, the slides were mounted with mounting solution (VectaMount, Vector Laboratories, Newark, NJ, USA) and coverslips.

### 2.7. Glycosaminoglycan (GAG) Quantification

The pellets were digested using 1 mg/mL Proteinase K, 1 mM iodoacetamide, 10 µg/mL Pepstatin A in 50 mM Tris, 1 mM EDTA buffer (250 μL) (pH 7.6; all Sigma-Aldrich, St. Louis, MO, USA) for 16 h at 56 °C, followed by Proteinase K inactivation at 100 °C for 10 min. To determine the amount of DNA, the cell lysates were treated with 0.415 IU/mL heparin and 1.25 µg/mL RNase for 30 min at 37 °C, followed by addition of 0.375 µL CYQUANT GR solution (ThermoFisher, Waltham, MA, USA). The samples were analysed using a SpectraMax Gemini plate reader with an excitation of 480 nm and an emission of 520 nm. As a standard, DNA sodium salt from calf thymus (Sigma-Aldrich, St. Louis, MO, USA) was used. To determine the amount of GAG, the cell lysates were diluted in PBS supplemented with 10 mM EDTA (pH 6.5) to a volume of 50µL and mixed with 200 µL of 32 mg/L 1,9-dimethylmethylene blue (DMB, Sigma-Aldrich, St. Louis, MO, USA) in 0.04 M Glycin, 0.04 M NaCl pH 3.0. Then the absorbance was measured on a Versamax microplate reader at 590 nm and 530 nm. A 530:590 nm ratio was used to determine the glycosaminoglycan concentration. As a standard, chondroitin sulphate sodium salt from shark cartilage (Sigma-Aldrich, St. Louis, MO, USA) was used.

### 2.8. Gene Expression Analysis

The pellets were manually homogenised with a pestle in 350 μL RNAstat (Tel-Test, Inc., Alvin, USA). Then, 70 μL chloroform (Sigma-Aldrich, St. Louis, MO, USA) was added and thoroughly mixed. Following a 10 min incubation at room temperature and phase separation at 10,000× *g* for 15 min, the aqueous phase was collected, mixed with an equal volume of 70% *v*/*v* ethanol and loaded onto an RNeasy^®^ micro kit column (Qiagen, Hilden, Germany). RNA was isolated and purified following manufacturer’s instructions. cDNA was reverse transcribed as per the manufacturer’s instructions, using the First Strand cDNA Synthesis Kit (Thermo Fisher, Waltham, MA, USA). The expression of the genes of interest was quantified using qPCR with a Bio-Rad CFX96 Real-Time PCR detection system (Bio-Rad), with either TAQman (ThermoFisher, Waltham, MA, USA) or SYBR-green-based chemistry (ThermoFisher, Waltham, MA, USA). The target genes were *COL2A1* (forward: 5′-GGCAATAGCAGGTTCACGTACA-3′, reverse: 5′-CGATAACAGTCTTGCCCCACT T-3′, and FAM-TAMRA-*COL2A1*-probe: 5′-CCGGTATGTTTCGTGCAGCCATCCT-3′), *COL10A1* (forward: 5′-CAAGGCACCATCTCCAGGAA-3′, reverse: 5′-AAAGGGTATTTGTGGCAGCATATT-3′, and FAM-TAMRA-*COL10A1*-probe: 5′-TCCAGCACGCAGAATCCATCTGA-3′), *MMP13* (forward: 5′-AAGGAGCA TGGCGACTTCT-3′, reverse: 5′-TGGCCCAGGAGGAAAAGC-3′, and FAM-TAMRA-*MMP13* probe: 5′-CCCTCTGGCCTGCGGCTCA-3′), *VEGFA* (forward: 5′-CTTGCCTTGCTGCTCTACC-3′, reverse: 5′-CACACAGGATGGCTTGAAG-3′), *RUNX2* (forward: 5′-ACGTCCCCGTCCATCCA-3′, reverse: 5′-TGGCAGTGTCATCATCTGAAATG-3′, and FAM-TAMRA- *RUNX2*-probe: 5′- ACTGGGCTTCTTGCCATCACCGA-3′), and *COL1A1* (forward: 5′-CAGCCGCTTCACCTACAGC-3′, reverse: 5′-TTTTGTATTCAATCACTGTCTTGCC-3′, and FAM-TAMRA-*COL1A1*-probe: 5′- CCGGTGTGACTCGTGCAGCCATC-3′). *B2M* (forward: 5′-TGCTCGCGCTACTCTCTCTTT-3′, reverse: 5′-TCTGCTGGATGACGTGAGTAAAC-3′) was selected for normalisation after evaluation of 3 different housekeeping genes. The expression data were analysed by the 2^−ΔCT^ method.

### 2.9. Cell Proliferation Analysis

To measure HUVEC proliferation, 2.5 × 10^3^ cells/cm^2^ were seeded in 48-well plates in 300 μL EGM2. After 24 h, the medium was replaced with a mix of 150 μL EBM and 150 μL pellet-derived CM (1:1). Non-conditioned medium + EBM was used as negative control. 10 mM dEdU (BaseClick GmbH, Munich, Germany) was added to label the DNA of replicating cells. After 24 h, the cells were washed three times with PBS and fixed with 4% formalin for 5 min. The EdU label was then revealed according the protocol provided by the manufacturer. The cells were counterstained with DAPI and imaged using fluorescence microscopy. Utilising the particle analysis macro in ImageJ, we determined the amount of positively stained cells for DAPI and EdU, and the percentage of EdU-positive cells was calculated. Three non-overlapping pictures/wells (1388 × 1040 μm^2^ per field) were taken, with triplicate wells for each independent experiment (N = 3 hMSC donors).

### 2.10. Cell Migration Analysis

Migration assays were performed by seeding HUVEC (1.5 × 10^5^ cells/cm^2^) in 24-well Transwell inserts (8 um pore size, Corning Life Sciences, Tewksbury, USA) in 200 µL endothelial basal medium (EBM) containing 0.05% BSA (Sigma-Aldrich, St. Louis, MO, USA). The CM from pellets was mixed with EBM at the ratio of 1:1 (total volume = 500 µL) and placed in the lower compartment of the wells. Non-conditioned medium + EBM was used as negative control. After 10 h of incubation at 37 °C 5% CO_2_, the cells on the membrane were fixed with 4% formalin/PBS, and the non-migrated cells from the upper surface of the membrane were removed with a cotton swab. The migrated cells on the lower surface of the membrane were stained with DAPI and then quantified by fluorescence microscopy and image analysis through ImageJ (software version 1.53t). Five non-overlapping pictures/wells were taken, with triplicate wells for each independent experiment. (N = 3 hMSC donors).

### 2.11. VEGFA ELISA

To quantify the amount of the pro-angiogenic cytokine VEGFA released into the CM, an ELISA assay (R&D System, Minneapolis, MN, USA) was performed. The capture antibody was diluted according to the manufacturer’s instructions and added into a 96-well plate at room temperature overnight. Then, wells were washed with 300 μL wash buffer two times and subsequently incubated with 300 μL reagent diluent at RT for 1 h. After washing the wells with wash buffer, CM from the pellets was added and the plate was sealed and incubated for 2h at RT. Then, 100 μL of detection antibody, 100 μL of working dilution of Streptavidin-HRP, 100 μL of substrate solution and 50 μL of stop solution were added into each well and incubated for 2 h, 2 h, 20 min, and 20 min, respectively. Between incubations, wells were washed with wash buffer twice, and the plate was sealed with an adhesive strip without exposure to direct light. Finally, the optical density of each well was measured at 450 nm and 570 nm (Versamax, Molecular Devices, San Jose, CA USA).

### 2.12. D Vascular Network Formation Assay in Fibrin Hydrogel

For each hydrogel, 300,000 HUVECs and 300,000 ASCs were resuspended in 50 µL of 20 mg/mL fibrinogen in 0.9% *w*/*v* NaCl (plasminogen, vWF and fibronectin-depleted human fibrinogen, MILAN Analytica AG). Human thrombin (Sigma-Aldrich, St. Louis, MO, USA) and Factor XIII (CSL Behring, King of Prussia, PA, USA) were added at a concentration of 6 U/mL to 50 µL of 40 mM CaCl_2_ solution. The cell suspension and enzyme solution were mixed to generate fibrin hydrogel with a total volume of 100 µL. 15–20 min was required for hydrogel crosslinking (37 °C). 270 µL of culture medium (1:1 mix of EGM2 and pellet CM from donor 1) was added to each hydrogel. Non-conditioned medium + EGM2 was used as negative control. The medium was renewed twice a week, and cultures were kept in a humidified atmosphere at 37 °C and 5% CO_2_. The hydrogels were fixed on day 14 of culture for confocal imaging analysis of vessel formation.

To perform whole-mount staining, the fibrin hydrogels were fixed with 1% paraformaldehyde (PFA) overnight at 4 °C. Samples were placed on a shaker to improve hydrogel penetration during staining. The gels were washed three times with PBS, for 3 h at 4 °C. After washing, 3% BSA and 5% donkey serum (Sigma-Aldrich, St. Louis, MO, USA) in PBS was used overnight to block non-specific binding. The gels were subsequently incubated with 1:100 anti-human laminin antibody (0.7 mg/mL stock, Abcam, Cambridge, UK) in PBS with 3% BSA and 5% donkey serum (Sigma-Aldrich, St. Louis, MO, USA) overnight. The PBS washing step was repeated, and incubation with 1:200 Alexa fluor 647 secondary antibody (2 mg/mL stock, Thermo Fisher, Waltham, MA, USA) at 4 °C was performed overnight with foil cover to avoid exposure to direct light. The hydrogels were imaged using a Leica Stellaris 5 low-incidence angle upright microscope, with an excitation wavelength of 638nm and a long-pass emission filter for laminin signal. The gels were optically scanned (distance between every scanned layer: 6 μm), and all the images were stacked through a Z-stack program in ImageJ for Vessel length density (VLD) analysis. Vessel lengths were measured by overlaying captured microscopic images with a square grid (field size = 200,000 μm^2^). Squares were randomly chosen and the length of each vessel (if any) in the selected squares was measured and summed up. For each sample, 10 fields of the whole image for vessel length measurements were obtained, with triplicate samples for each experimental group. A schematic of the VLD measurement is reported in Appendix A.

### 2.13. Statistical Analysis

Data representation was performed using GraphPad Prism (software version 8.0), and statistical analyses was performed with SPSS 24 (IBM). The normality of the data was first verified using the Kolmogorov–Smirnov test. Subsequently, for all multiple comparisons, a linear mixed model with Bonferroni correction was used; the different conditions were considered as a fixed parameter and the donor as a random factor. Statistical significance was evaluated between conditions within the same time-point and defined as *p* < 0.05. All results are presented as mean ± standard deviation (SD).

**Figure 1 cells-12-01202-f001:**
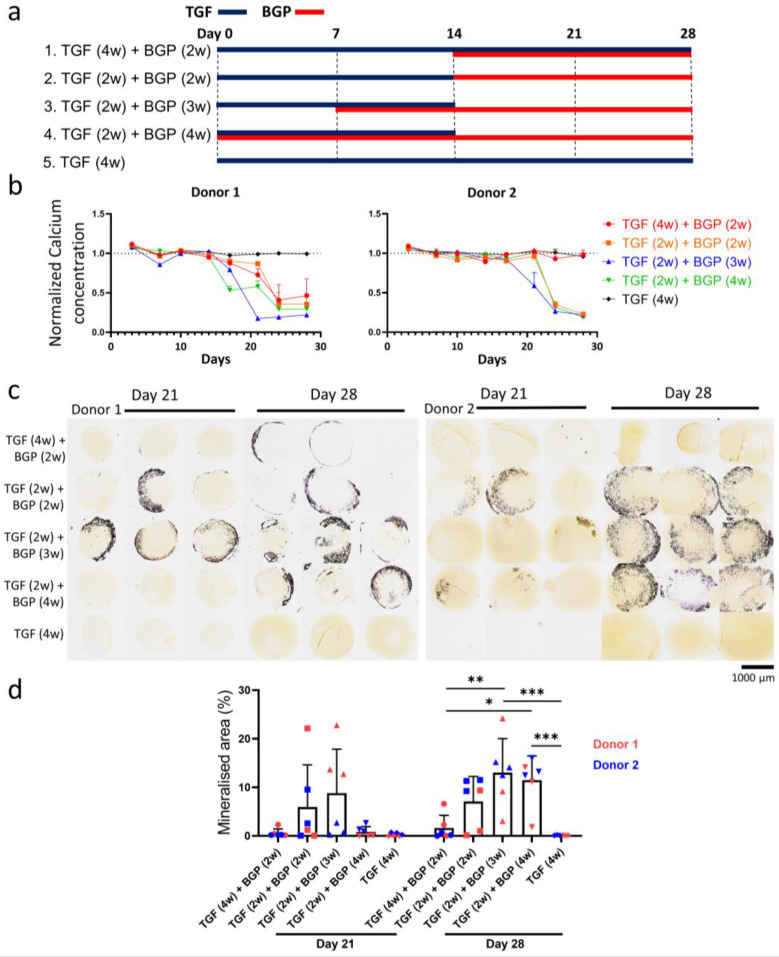
Sequential exposure to TGF-β and BGP leads to in vitro mineralisation of chondrogenic pellets. (**a**) Culture scheme of the different approaches to stimulate in vitro pellet mineralisation over 28 days. N = 2 hMSC donors in triplicate samples per time-point. (**b**) Longitudinal measurements of extracellular calcium levels in the culture medium to assess calcium uptake by the pellets over 28 days. Data were normalised vs. medium only and shown as average ± SD. Error bars denote standard deviation. (**c**) Von Kossa-stained histological sections of the pellets subjected to the different mineralisation protocols (3 pellets/condition from 2 hMSC donors on day 21 and 28 of culture). (**d**) Quantification of the mineralised area using Von Kossa-stained histological sections. Data are presented as average ± SD (N = 2 hMSC donors). * 0.01 < *p* < 0.05, ** 0.01 < *p* < 0.001, *** *p* < 0.001.

## 3. Results

### 3.1. Establishment of the Culture Protocol to Induce In Vitro Mineralisation of Chondrogenic Pellets

In order to establish an effective culture scheme to induce in vitro mineralisation of hMSC-derived chondrogenic pellets, we tested several combinations of TGF-β and BGP addition over 28 days of culture (Figure 1a). For all experimental groups, TGF-β was added from day 0 of pellet culture to induce chondrogenesis. In the case of group 1 (TGF (4w) + BGP (2w)), we added BGP to the chondrogenic medium from day 14, according to a protocol that was previously used by our group for pellet mineralisation [8]. Since we generally observed that with this strategy the onset of mineralisation does not always occur within 28 days (Figure 1b,c; donor 2), we hypothesised that a briefer chondrogenic priming whereby TGF-β is removed on day 14 could facilitate BGP-induced mineralisation. Hence, for groups 2, 3 and 4, TGF-β administrations were reduced to 2 weeks and BGP was added for 2, 3 or 4 weeks, respectively (TGF (2w) + BGP (2w/3w/4w)). The media from the pellets were collected at different time points to monitor the drop in extracellular calcium concentration, which is indicative of calcium uptake during mineralisation (Figure 1b). Group 1 showed a decline in extracellular calcium levels during week 3 in the case of donor 1, but no sign of calcium uptake was observed in the case of donor 2 within 28 days. Interestingly, groups 2–4 all showed a decline in calcium levels, suggesting an increase in the uptake of calcium by the pellets, within 4 weeks of culture for both donors. Of note, group 3 showed more consistency overall in terms of early onset of mineralisation among the two donors.

Von Kossa staining of the pellets at day 21 and 28 indicated very limited (donor 1) to no mineralisation (donor 2) for the TGF (4w) +BGP (2w) group within 28 days (Figure 1c,d). Groups 2–4 showed an overall larger mineralised region, mainly in the peripheral area of the pellets. In agreement with the calcium uptake results, we found slightly accelerated or more intense deposition of minerals in the case of the group 3. Thus, the culture scheme of group 3 (TGF (2w) + BGP (3w)) was selected for all subsequent experiments.

### 3.2. BGP-Driven Mineralisation Induces Cell Death and Matrix Changes in Chondrogenic Pellets

We next investigated how mineralisation of chondrogenic pellets affects cell viability and matrix components. Importantly, these aspects may significantly impact the pro-angiogenic potential of cartilage. To answer these questions, we subjected pellets from 3 hMSC donors to the optimised mineralisation protocol and performed von Kossa staining, thionine staining, DMB assay and Live/Dead FACS assay throughout differentiation. To discriminate the effect of BGP-induced mineralisation from the effect of TGF-β withdrawal, control pellets were cultured with TGF-β for 2 weeks, and the factor was then removed from the medium for the last 2 weeks without BGP addition (TGF (2w) condition) (Figure 2a). As expected, calcium uptake measurements and von Kossa staining showed that BGP addition induced consistent pellet mineralisation (Figure 2b–d; Appendix A), starting from week 2–3 of culture for all donors. In the case of donors 3 and 4, the presence of mineral deposits in the pellets was detected from day 14, and further progressed afterwards; donor 5 showed slower and more limited mineralisation (Figure 2c,d). H&E staining of the pellets further evidenced hypertrophic chondrocytes embedded in the calcified matrix of BGP-treated pellets (Appendix A).

Thionine staining of the pellets and a DMB assay performed on the pellet lysates showed that glycosaminoglycans (GAGs) accumulated over time in the matrix of chondrogenic pellets (TGF (4w)), and this was not significantly affected by the removal of TGF-β after 2 weeks of culture (TGF (2w)) (Figure 2e,f). In the case of mineralised pellets (TGF (2w) + BGP), the GAG content was significantly decreased at all time-points (Figure 2f). During endochondral ossification, most hypertrophic chondrocytes in mineralised cartilage undergo apoptosis. Hence, we next aimed to quantify the number of viable cells in mineralised pellets. The live/dead FACS assay showed that on day 14, chondrogenic and mineralised pellets exhibited a low % of dead cells (TGF (4w): 5.69 ± 4.21%; TGF (2w) + BGP: 9.58 ± 7.93%). On day 28, mineralised pellets exhibited an increased ratio of dead cells (TGF (2w) + BGP: 34.73 ± 8.08%) when compared to chondrogenic pellets (TGF (4w): 8.47 ± 1.00%; TGF (2w): 7.29 ± 0.58%) (Figure 2g, bar graph).

To further investigate changes in the cartilage matrix, immunohistochemistry and gene expression analysis for collagen type II and X were performed. TGF-β induced abundant collagen type II and X production and accumulation during chondrogenesis, which was maintained in the last 2 weeks of culture regardless of the presence of TGF-β in the culture medium (Figure 3a,b). Mineralised pellets did not exhibit major qualitative differences in collagen type II or X staining when compared with the other conditions. Further gene expression analyses showed that the mRNA expression of *COL2A1* and *COL10A1* was significantly reduced upon BGP-induced mineralisation, particularly at day 21 and 28 of culture (Figure 3c,d). A similar trend was observed for the osteogenic markers RUNX2 and COL1A1 (Appendix A).

In summary, our data show that BGP-induced mineralisation leads to specific changes in chondrogenic pellets, namely increased cell death, loss of GAGs and reduced expression but not protein abundance of collagen type II and X.

### 3.3. In Vitro Mineralisation Reduces the Pro-Angiogenic Potential of Chondrogenic Pellets

To gain insight into how cartilage mineralisation may potentially affect vessel attraction and invasion, we next performed multiple experiments to examine the angiogenic potential of in vitro mineralised pellets. First, we quantified *MMP13* and *VEGFA* mRNA expression levels in hMSC-derived chondrocytes during in vitro mineralisation due to the relevance of these factors to matrix remodelling and angiogenesis during endochondral ossification. The TGF (2w) and TGF (2w) + BGP groups exhibited significantly lower *VEGFA* and *MMP13* expression on day 21 (*VEGFA*) and 28 (*VEGFA*, *MMP13*) when compared to TGF (4w), suggesting that TGF-β supplementation is necessary to sustain their expression (Figure 4a). While TGF-β withdrawal caused a significant decrease in VEGFA release in the medium, the effect was further anticipated and exacerbated by BGP-induced mineralisation (Figure 4b). Next, endothelial cell migration and proliferation assays were performed to investigate how HUVEC behaviour was affected by the CM from the pellets. The withdrawal of TGF-β from chondrogenic pellets (TGF (2w) group) led to decreased HUVEC migration in comparison to the TGF (4w) group on day 28 (Figure 4c). In the case of mineralised pellets (TGF (2w) + BGP group), an inhibitory effect on cell migration was already observed for CM derived from day 21 pellets. Regarding HUVEC proliferation, there was no effect due to the withdrawal of TGF-β, while BGP-induced mineralisation resulted in significantly decreased proliferation levels compared with the other groups on day 21 and 28 (Figure 4d). Finally, to further validate the overall reduction in the pro-angiogenic potential of mineralised cartilage, we applied a model of 3D vascular network formation in fibrin hydrogels in the presence of CM. Here, we compared the effect of CM from TGF (4w) and TGF (2w) + BGP pellets at day 14 and day 28 on vessel length density (VLD) (Figure 4e). In the case of day 14 CM, we observed similar VLDs for TGF (4w) and TGF (2w) + BGP groups (17.86 ± 0.38 and 17.52 ± 0.23 mm/mm^2^, respectively). However, hydrogels cultured with CM derived from day 28 mineralised pellets exhibited lower VLD when compared to the day 28 TGF (4w) group (19.12 ± 0.35 and 13.41 ± 1.23 mm/mm^2^, respectively) (Figure 4e).

In conclusion, our data show that the ability of chondrogenic pellets to induce endothelial cell migration, proliferation and tube formation is strongly reduced upon prolonged cartilage mineralisation.

## 4. Discussion

In this study, we investigated the changes in angiogenic potential that occur during the mineralisation of hMSC-derived cartilage. While the pro-angiogenic potential of tissue-engineered cartilage is well documented [4,31,32], the manner in which mineralisation specifically affects angiogenesis still requires investigation. Here, we optimised an in vitro strategy to induce the mineralisation of hMSC-derived chondrogenic pellets and characterised the cellular and matrix changes that occur overtime. Our data show that cartilage mineralisation leads to a strong reduction in the ability of tissue-engineered cartilage to stimulate migration, proliferation, and network formation by endothelial cells. These findings suggest that mineralisation itself is not one of the major factors in driving blood vessels to infiltrate cartilage during endochondral ossification, but may instead significantly delay vascularisation processes, at least in vitro.

We first focused on establishing a reliable culture method to introduce mineralisation in chondrogenic pellets. Previously published protocols for in vitro mineralisation often require the use of multiple culture media and/or prolonged culture periods (4–6 weeks) due to the induction of mineralisation after prolonged maturation of the chondrogenic template [4,6,8,10,37,38,39]. Since there is evidence that TGF-β can inhibit mineralisation [40,41,42], we here withdrew TGF-β from the pellets after 2 weeks of culture, and showed that this does not negatively affect the cartilage matrix or the expression of matrix-related genes.

Furthermore, previous studies from our group have shown that 7 days of chondrogenic priming are sufficient to achieve cartilage mineralisation and bone formation after subcutaneous implantation of the pellets [9]. Therefore, we started adding BGP from day 7 of chondrogenesis. We succeeded in inducing the mineralisation of hMSC pellets as early as day 14. Interestingly, the region of mineralisation normally originates from the edge of pellets in our studies. This is consistent with previous in vitro mineralisation studies and may to some extent mimic the formation of the bone collar on the border of the cartilage rudiment during endochondral ossification [43]. These cells produce ECM to form the osteoid structure and also aggregate calcium phosphate to form hydroxyapatite [44]. Previously, we have observed that normally the same peripheral area of chondrogenic pellets undergoes bone formation after in vivo transplantation, leading to the generation of a ring of bone tissue [45]. The reason for this spatial difference in the distribution of mineral deposits and bone formation is not fully understood. Some possible explanations are the gradient in the diffusion of nutrients, growth factors and oxygen, as well as intrinsic heterogeneity in cellular populations within the pellets. Application of spatial omics approaches to these experimental models may provide further insights into the cellular processes underlining cartilage mineralisation and bone formation during endochondral ossification.

In vitro cartilage mineralisation led to specific changes in matrix components. While collagen II and X protein expression was well maintained overall, the mRNA expression of collagens and RUNX2 was reduced after prolonged mineralisation. This is not surprising, since hypertrophic chondrocytes generally exhibit high expression of osteogenic markers, even in the absence of active mineralisation. The decrease in the expression of genes encoding for matrix and osteogenic factors may be due to an overall transition from an anabolic (chondrogenic) stage to a “remodelling” stage, which is consistent with the fact that in vivo mineralised cartilage is infiltrated by blood vessels and gradually resorbed. At this stage, downregulation in the expression of these genes may occur. Alternatively, the downregulation of these factors may be related to the limitation of in vitro models, wherein only a limited number of cell types can be included and not all cellular crosstalk can be captured. BGP-induced mineralisation further led to a strong decrease in GAGs in the matrix from day 21 of culture. The fact that chondrogenic pellets from which TGF-β was withdrawn after 14 days did not show GAG loss indicates that this process was related to BGP-driven mineralisation. During the early stages of endochondral ossification, GAGs are abundantly present in the matrix around hypertrophic chondrocytes. It has previously been shown that negatively charged GAGs could inhibit hydroxyapatite formation [46,47,48,49], and the cartilage matrix thus needs to be degraded for endochondral ossification to proceed. With the expansion of the mineralisation border, the ECM containing collagen I will be calcified. Meanwhile, the protein components of the proteoglycans will be finally degraded by enzymes such as matrix metalloproteases (MMPs) and aggrecanases produced by the hypertrophic chondrocytes themselves [50]. Viable chondrocytes also play a direct role in maintaining GAGs [51] and the overall matrix integrity. In this study, we found that high levels of cell death occurred at the late time-point of in vitro mineralisation (day 28). While few hypertrophic chondrocytes can transdifferentiate into osteoblasts during endochondral ossification, most late hypertrophic chondrocytes will undergo apoptosis and be removed from the cartilage template [52]. The increased levels of Pi and Ca^2+^ at the mineralisation front could locally induce chondrocyte apoptosis. Mansfield et al. showed that this ion pair can induce chondrocyte death by causing an increase in reactive oxygen species (ROS), which can initiate the induction of apoptosis [53]. Accordingly, studies on rickets, a disorder associated with failure of endochondral ossification and impaired mineralisation, have shown that hypophosphatemia leads to enlarged late hypertrophic chondrocytes and diminished cell death due to reduced caspase activation in the presence of lower Pi levels [54,55]. Whether and to what extent these or other mechanisms mediate chondrocyte death during in vitro mineralisation remains to be clarified.

Prolonged in vitro mineralisation strongly diminished the pro-angiogenic potential of hMSC-derived cartilage, as demonstrated by the proliferation and migration assay performed with HUVECs exposed to conditioned media from the pellets. The withdrawal of TGF-β itself led to a drop in the production of VEGFA by the pellets and a reduction in their ability to stimulate HUVEC migration. TGF-β has a variety of biophysical effects on MSCs [56,57], and it was reported that angiogenic factors, such as hepatocyte growth factor (HGF), angiogenin (ANG) or VEGFA, can be released from MSCs exposed to TGF-β [31,32,45,57]. BGP-driven mineralisation further exacerbated this loss of pro-angiogenic potential. It is likely that cell death occurring after prolonged mineralisation contributed to the reduction in VEGFA secretion. The reduction of VEGFA in (normalised) gene expression levels also indicates reduced VEGFA expression in each cell. Furthermore, TGF (2w) pellets did not show increased cell death, but exhibited decreased VEGF expression and secretion. In conclusion, it is likely that both cell death and cellular downregulation of angiocrine expression and secretion contributed to the reduced stimulatory effect of mineralised cartilage on endothelial cell migration and proliferation.

When we performed a 3D vascular network formation assay in fibrin hydrogel, no significant difference in vascular network formation was found in the case of the conditioned medium from chondrogenic and mineralised pellets on day 14. Since the VEGFA level already dropped on day 14 in the presence of BGP, it is likely that additional secreted factors may still be able to support vessel network formation. Nevertheless, the pro-angiogenic effect was again lost after prolonged mineralisation. While our data clearly show a negative impact of mineralisation on endothelial cell migration, proliferation and tube formation, further research should aim to untangle the exact mechanisms by which secreted factors from (mineralised) cartilage regulate endothelial cell differentiation and tube formation. Furthermore, it is important to consider that the use of a conditioned medium can only recapitulate unidirectional cellular interactions mediated by secreted factors, and future studies will be necessary to explore the additional mechanisms by which mineralisation can impact angiogenesis. In this work, we focused our attention on the very initial stages of angiogenesis during endochondral ossification, whereby new vessels form and grow towards a mineralising cartilage template. The later infiltration of blood vessels into the cartilage template with the formation of a vascularised marrow niche is likely heavily dependent on local and direct cell contact [58,59]. To completely address these phenomena, experimental models with direct contact between the matrix and forming vessels will be needed, so that the processes of matrix remodelling and vascular infiltration can be further studied.

Tissue-engineered grafts using MSCs hold a great deal of promise for bone regeneration, but achieving sufficient graft vascularisation is still a concern. While there is abundant evidence that such an issue can be overcome by inducing bone formation via the endochondral pathway, translating this approach to patients will require addressing the issues of prolonged in vitro handling and relatively slow in vivo bone formation. In recent years, extensive research efforts have been directed towards developing tools to minimise the time required for priming the cells and achieving graft vascularisation after implantation. Nevertheless, ideal strategies that can at the same time support rapid angiogenesis and MSC-mediated tissue production are still under investigation. The results of our study show that cartilage mineralisation in the tissue engineering setting may be a crucial process counteracting the pro-angiogenic potential of cartilage. Strategies attempting pre-mineralisation of the chondrogenic constructs (prior to implantation) should be handled with care. Whether it is possible to “engineer” the mineralisation process by providing further cues that can sustain angiogenesis is a relevant question for the future. In this regard, suitable in vitro models to answer this and other fundamental questions on the processes of endochondral bone formation are still lacking. Several co-culture systems with MSCs and endothelial cells or endothelial progenitor cells (EPCs) have been established [20], and these models can be applied to study how cellular interactions can affect specific osteogenic and angiogenic properties. However, further bone or vascular development of the tissue-engineered constructs still has to be achieved with in vivo transplantation [60,61,62], which poses challenges related to the controllability of the models. Particularly, to recapitulate the processes of cartilage remodelling, mineralisation and angiogenesis in a coordinated manner in vitro, more complicated models are needed. This not only relates to the process of blood vessel formation and physical interactions with the mineralised matrix, but also to the presence of additional cell types that may directly intervene in these processes, such as osteoclasts and macrophages. The outcomes of our study provide relevant knowledge for the development of such models, in particular concerning the coordination of cartilage formation, mineralisation and angiogenesis.

## 5. Conclusions

In conclusion, we established a reliable strategy to induce mineralisation of hMSC-derived cartilage in vitro and characterised tissue changes on both a cellular and histological level. We showed that mineralisation leads to a progressive decrease in the ability of tissue-engineered cartilage to stimulate endothelial cell migration, proliferation and tube formation. These data indicate that the pro-angiogenic potential of hypertrophic and mineralising cartilage may be strictly stage-dependent, and these features represent a crucial aspect to consider in the design of bone tissue-engineering strategies. Finally, our study can act as the basis for development of more sophisticated in vitro models to recapitulate and study the vascularisation of mineralised cartilage during endochondral ossification.

## Figures and Tables

**Figure 2 cells-12-01202-f002:**
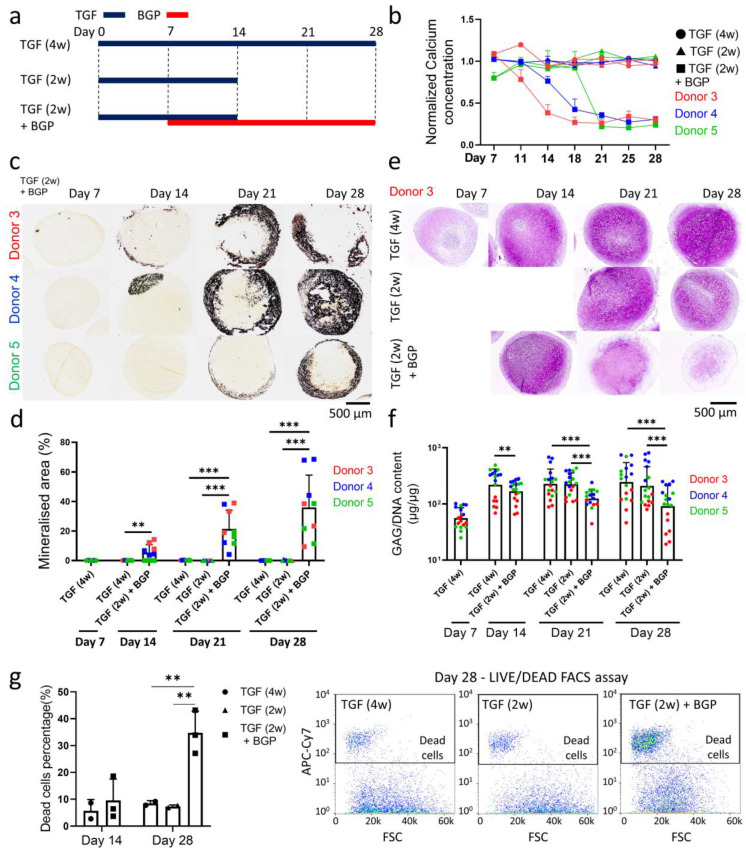
Effect of BGP-induced mineralisation on GAG production and cell viability. (**a**) Culture scheme of chondrogenic (TGF-β (4w)) and mineralised pellets (TGF-β (2w) + BGP). A third group whereby TGF-β is withdrawn on day 14 but BGP is not added was included to account for potential effects caused by TGF-β withdrawal and unrelated to BGP. (**b**) Longitudinal measurements of extracellular calcium levels in the culture medium to assess calcium uptake by the pellets over 28 days. Data are normalised vs. medium only and shown as average ± SD. (**c**) Von Kossa-stained histological sections of the pellets of the BGP-treated group. Groups treated with TGF-β only did not exhibit any staining (Appendix A). (**d**) Quantification of the mineralised area using Von Kossa-stained histological sections. Data are shown as average ± SD (N = 3 hMSC donors). (**e**) Representative images of the thionine staining of the pellets for the different experimental groups for donor 3. (**f**) Quantification of GAG content in pellet lysates by DMB assay. Data are shown as average ± SD (N = 3 hMSC donors). (**g**) Flow cytometry-based live/dead assay of the pellets on day 14 and day 28 of culture. Data in the bar graph indicate the % of dead cells in the pellets and are shown as average ± SD (2–3 replicates per group per time point). Representative FACS plots are reported on the right. ** 0.01 < *p* < 0.001, *** *p* < 0.001.

**Figure 3 cells-12-01202-f003:**
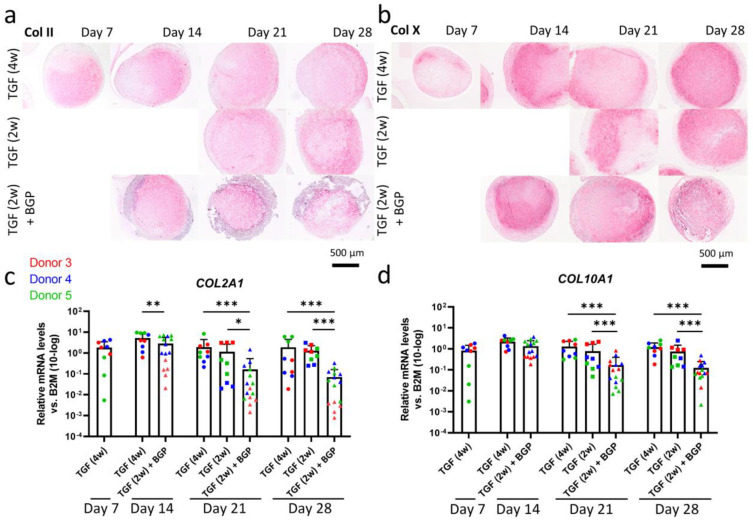
Effect of in vitro mineralisation on collagen expression and accumulation in the pellets. Collagen type II (Col II) (**a**) and collagen type X (Col X) (**b**) immunohistochemical staining of the pellets. Representative images are shown. *COL2A1* (**c**) and *COL10A1* (**d**) mRNA expression determined by qRT-PCR. *B2M* was used as the housekeeper gene. Data are presented as average ± SD (N = 3 hMSC donors). * 0.01 < *p* < 0.05, ** 0.01 < *p* < 0.001, *** *p* < 0.001.

**Figure 4 cells-12-01202-f004:**
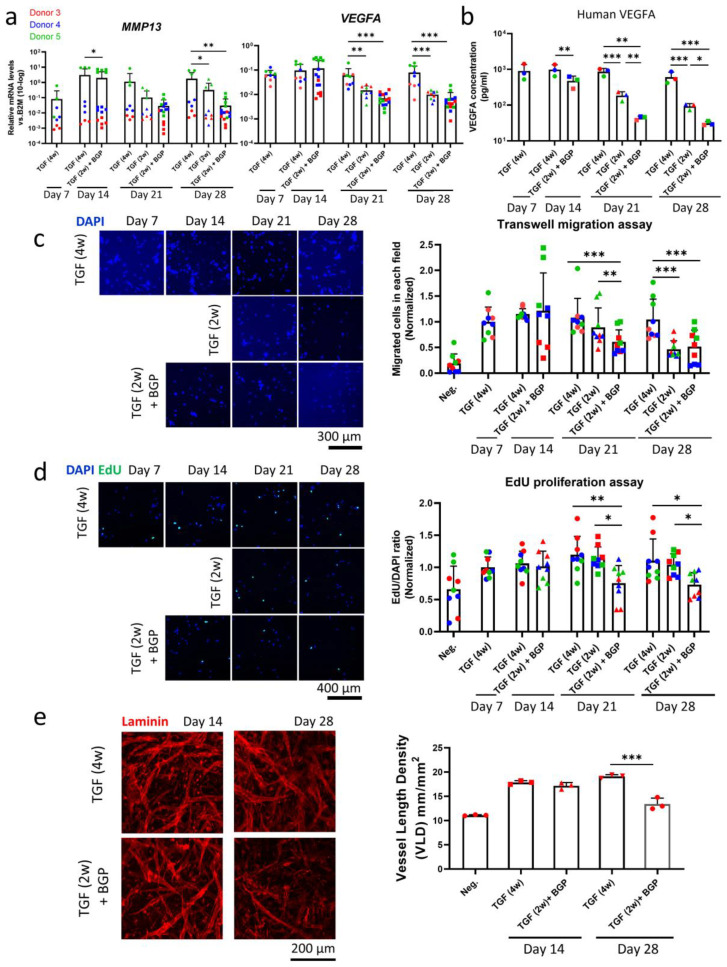
Assessment of the angiogenic potential of in vitro mineralised pellets. (**a**) *MMP13* and *VEGFA* mRNA expression was determined by qRT-PCR. *B2M* was used as the housekeeper gene (N = 3 hMSC donors). (**b**) ELISA-based quantification of secreted VEGFA in the CM of chondrogenic and mineralised pellets over time (N = 3 hMSC donors). (**c**) Transwell migration assay performed with HUVECs exposed to the CM of the pellets. Representative images of migrated cells are shown. The graph displays the normalised number of migrated cells per field (vs. TGF (4w) day 7) (N = 3 hMSC donors). (**d**) dEdU proliferation assay performed with HUVECs exposed to the CM of the pellets. The graph displays the normalised EdU-positive cells per field (vs. TGF (4w) day 7) (N = 3 hMSC donors). (**e**) 3D tube formation assay in fibrin hydrogel performed using the CM from chondrogenic (TGF (4w)) and mineralised (TGF (2w) + BGP) groups from donor 3. Representative z-stack images obtained by confocal microscopy are shown. The graph displays the average VLD values per condition. All data are presented as average ± SD. * 0.01 < *p* < 0.05, ** 0.001 < *p* < 0.01, *** *p* < 0.001.

## Data Availability

All raw data related to this study are included as a Appendix A.

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
