# Peer review of "In Vitro Mineralisation of Tissue-Engineered Cartilage Reduces Endothelial Cell Migration, Proliferation and Tube Formation"

_cells, 2023, doi:10.3390/cells12081202_

Round 1

Reviewer 1 Report

Based on the research hypothesis and title of the manuscript, the authors should perform more experiments i.e., angiogenic differentiation of ECs in vitro using CM and analyze the mRNA and protein level expression of angiogenic differentiation markers in ECs.  

Reviewer 2 Report

This study investigates how cartilage mineralisation can affect vascularization. MSCs-derived chondrogenic pellet are treated with β-glycerophosphate to generate in vitro mineralized cartilage. The authors also showed that conditioned medium from mineralised pellets has a reduced ability to stimulate endothelial cell migration, proliferation, and tube formation.

The study is well designed and conducted. The manuscript is well written, and the results are well organised. Methods are clearly presented, and introduction provides a sufficient background to the readers.

There are a few minor issues to be addressed.

1.    Methods: in “Cell culture” section (row 118), authors mention about Adipose-derived stromal cells (ASCs), but they do not mention for which purpose these cells were used. Are the pellets derived from ASCs or BMSCs? Some literature reports indicate that adipose-MSCs possess a delayed osteogenic capacity compared to BMSCs. This is a characteristic shared by the cells themselves (doi:10.1186/s13287-018-0914-1) and by their secretome (doi:10.1007/s00441-020-03315-5). There are evidences that these two different sources could lead to different outcomes in endochondral ossification. Adipose-derived MSCs are more prone to induce vascularization, while bone marrow-derived MSCs push ossification process more. Please check and clarify. Also, if the aim of the authors is to perform a comparison, this is not clearly explained and understandable. In addition, a single donor for ASCs is not sufficient to be statistically significant.

2.    A number of two maximum three replicates is used for all the experiments. Due to the variability of primary cell cultures, an increase to 5 replicates would be preferable, to improve the robustness of presented data. If this is not achievable, please justify.

3.    Figure 2: different pellets in this figure are named as donor 3, 4 and 5, but in the text the authors refer to them as donor 1, 2 and 3 (rows 355-357). Please check.

4.    Figure 2c: authors show Von Kossa staining only related to mineralised pellets (TGFbeta 2w + BGP). Are they the same pellets used for protocol optimization (Figure 1)? Since they are supposed to be different donors, please provide Von Kossa staining of TGFbeta (2w) and TGFbeta (4w) also for donor 3, 4 and 5, as supplemental material.

5.    Figure 2d: GAG accumulation by thionine staining is showed. If possible, a quantitative analysis (i.e., 1-9-dimethylmethylene blue assay) is suggested.

6.    Figure 3: “Mineralised pellets did not exhibit major differences in collagen type II or X staining when compared with the other conditions. Gene expression analyses showed that COL2A1 and COL10A1 expression was significantly reduced upon BGP-induced mineralisation, particularly at day 21 and 28 of culture” Do the authors have more insight into the possible molecular mechanism involved? Is it possible to determine the expression of specific mineralization-related factors in this model (i.e., Runx2, BMP2, Col1a1)?”

Finally, haematoxylin/eosin staining of pellet should be provided, with high magnification pictures, to show differences in cell morphology, between different treatments.

Reviewer 3 Report

This manuscript introduces an in vitro platform to optimally induce mineralization from human mesenchymal stromal cells (hMSC)-derived chondrogenic pellets by treatment with transforming growth factor-β (TGF-β) and β-glycerophosphate (BGP). Conditioned media from those mineralized pellets showed reduced ability to stimulate human umbilical vein endothelial cells (HUVECs) migration, proliferation and tube formation in vitro. In general the problem is well motivated and there is clearly a strong need for consistent engineered platforms for mineralization of cartilage. Key issues to focus on is to add more robust quantification and statistical analysis, and expand the discussion to address the heterogeneity between donors and limitations of the current system on accessing angiogenesis. 

  1. In figure 1c, the mineralization seems to be quite heterogeneous in donor 1 TGF (2w)+ BGP (3w). The third pallet had less mineralization on day 28. It will be better to show some level of quantification and statistical analysis to address the variability within and between donors. Similarly, it should be possible to quantify the level of mineralization based on the images in figure 1c, 2c, and 3a as % mineralization area to show the changes across time and between samples which will strengthen the arguments in the manuscript. 

  2. The staining of collagen type II and X was not quantified and it will make the conclusion stronger if statistical analysis was performed, instead of just showing the images alone and describing them in the text as significant changes. 

  3. The variability of mineralization from each donor seems high. Donor 5 has a low mineralization and high MMP13, which seems to lead to significant changes in the pooled graph in fig 2 and 3 while donor 3 & 4 are pretty comparable. This seems to affect the statistical comparison in all graphs related.

  4. Given the significant amount of apoptotic cells after 28 days of mineralization, is the reduction of VEGF secretion in the medium changed due to less secretion from each cell or an overall reduction due to fewer cells? The mineralization process is associated with many biophysical properties that may directly influence angiogenesis but in this in vitro setting, only the condition medium was used to assess the effects. This may lead to generalization of mechanisms and conclusions. An expanded discussion will help address this issue. 

Round 2

Reviewer 1 Report

The revised manuscript is acceptable for publication.